# Nanosurgical Manipulation of Titin and Its M-Complex

**DOI:** 10.3390/nano12020178

**Published:** 2022-01-06

**Authors:** Dominik Sziklai, Judit Sallai, Zsombor Papp, Dalma Kellermayer, Zsolt Mártonfalvi, Ricardo H. Pires, Miklós S. Z. Kellermayer

**Affiliations:** 1Department of Biophysics and Radiation Biology, Semmelweis University, Tűzoltó Str 37-47, H1094 Budapest, Hungary; sziklaidominik09@gmail.com (D.S.); juditsallai@gmail.com (J.S.); pzsm32@gmail.com (Z.P.); dalmakeller@gmail.com (D.K.); falvizs@gmail.com (Z.M.); rhjpires@gmail.com (R.H.P.); 2Zentrum für Innovationskompetenz-Humorale Immunreaktionen bei Kardiovaskulären Erkrankungen, University of Greifswald, 17489 Greifswald, Germany

**Keywords:** atomic force microscopy, nanomanipulation, polymer extension, domain unfolding, mechanical stability, titin, molecular mechanics

## Abstract

Titin is a multifunctional filamentous protein anchored in the M-band, a hexagonally organized supramolecular lattice in the middle of the muscle sarcomere. Functionally, the M-band is a framework that cross-links myosin thick filaments, organizes associated proteins, and maintains sarcomeric symmetry via its structural and putative mechanical properties. Part of the M-band appears at the C-terminal end of isolated titin molecules in the form of a globular head, named here the “M-complex”, which also serves as the point of head-to-head attachment of titin. We used high-resolution atomic force microscopy and nanosurgical manipulation to investigate the topographical and internal structure and local mechanical properties of the M-complex and its associated titin molecules. We find that the M-complex is a stable structure that corresponds to the transverse unit of the M-band organized around the myosin thick filament. M-complexes may be interlinked into an M-complex array that reflects the local structural and mechanical status of the transversal M-band lattice. Local segments of titin and the M-complex could be nanosurgically manipulated to achieve extension and domain unfolding. Long threads could be pulled out of the M-complex, suggesting that it is a compact supramolecular reservoir of extensible filaments. Nanosurgery evoked an unexpected volume increment in the M-complex, which may be related to its function as a mechanical spacer. The M-complex thus displays both elastic and plastic properties which support the idea that the M-band may be involved in mechanical functions within the muscle sarcomere.

## 1. Introduction

Titin is a highly elastic multidomain [1] filamentous protein of striated muscle that spans the half sarcomere from the Z-disc to the M-band [2,3]. Its multidomain architecture makes it one of the most versatile proteins of the sarcomere [3]. Titin’s physical role is to align and stabilize the sarcomere and maintain its geometric symmetry [4]. Besides its mechanical functions, titin is also thought to be important in providing a blueprint for sarcomeric organization [5,6,7] and sensing the sarcomere’s contractile status [8,9,10]. By alternative splicing, many titin isoforms may be generated [11], which assist in the mechanical adaptation to different physiological and pathological challenges in striated muscle. Furthermore, titin has been suggested to contribute to active muscle force generation driven by rapid refolding of its globular domains [12,13,14]. Titin’s numerous functions explain the increasing recognition of its contribution to a number of diseases [2,4]. Structurally, titin is woven into the M-band in the middle of the sarcomere [15,16]. Purified titin samples display a globular head-like structure at the C-terminus, which is most likely composed of the proteins making up the M-band [17]. The M-band is a complex supramolecular assembly that assists in cross-linking and registering the thick filaments [15,18]. M-band structure has been remarkably well resolved in early electron microscopic experiments that revealed a set of mirror-symmetrically placed lines, called M-lines, within the M-band, and provided a structural model according to which the M-band is a hexagonal lattice traversed, in the center of the hexagons, by the myosin thick filaments [19,20]. The main protein contributing to the structure and function of the M-band is myomesin [21,22], but additional proteins such as M-protein, myomesin-3, muscle-type creatine kinase, obscurin, and others also play a role [15,18]. The structure, mechanics, and function of the M-band and its crucial components have been investigated by numerous approaches such as electron microscopy [19] and tomography [20], immunoelectron microscopy [21], x-ray crystallography [23], superresolution microscopy [24], force spectroscopy [25], and molecular dynamics simulations [26]. However, the exact arrangement of the molecular components in the M-band and the molecular mechanisms of their response to the mechanically ever-changing local environment are still largely unknown. Because the globular head of isolated titin molecules appears to reflect the in situ properties of the M-band, we considered it a worthy strategy to investigate the M-band by exploring the structural and mechanical features of this globular head with single-molecule manipulation methods. Such nanoscale manipulation methods can be used to deform, displace, cut and mechanically modify biomolecular or cellular objects with nanometer precision, and they have contributed to the emergence of the concept of nanosurgery [27]. In nanosurgery, fast lasers [28,29,30], optical tweezers [31], or atomic force microscopy (AFM) [32] are employed for nanoscale manipulation and modification. AFM has been used for the nanomanipulation and nanodissection of a wide range of biological objects such as DNA [32,33,34,35], chromosomes [36], fibers [37,38], cells [39], and proteins [40,41,42,43]. Thus, nanosurgery offers a novel approach to dissect and uncover the features of titin’s globular head, hence the structural unit of the M-band.

## 2. Materials and Methods

### 2.1. Titin Purification

Titin was purified from male New Zealand white rabbit *m. longissimus dorsi* by using published methods [44,45]. The procedure was approved (approval number: XIV-I-001/29-7/2012) by the Semmelweis University Regional and Institutional Committee of Science and Research Ethics and by the Directorate for Food-chain Safety and Animal Health of the Government of Pest County with reference to the Hungarian Law on the Protection and Humane Treatment of Animals (XXVIII/1998). In the last step of purification, 1.5 mL of titin-rich supernatant (OD_280_~18) was loaded on a high-aspect-ratio (length: 120 cm, diameter: 0.8 cm) Sepharose CL-2B column, eluted with chromatography buffer (30 mM K-phosphate, pH 7.0, 0.6 M KCl, 0.1 mM EGTA, 0.3 mM DTT, 2 µg/mL leupepetin, 1 µM E64, 0.01% NaN_3_, 0.05% Tween-20) at a flow rate of 0.2 mL/min, and collected into 1-mL fractions. Samples were stored on ice in the presence of protease inhibitors (40 μg/mL leupeptin, 20 μM E64) and used within three weeks. Purity was assessed by using vertical SDS agarose gel electrophoresis (0.8%) according to established methods [46].

### 2.2. Titin Sample Preparation for AFM

Samples were prepared for AFM imaging as described previously [47]. Briefly, protein samples (at a concentration between 0.3–0.5 mg/mL) were diluted (100×) in chromatography buffer, then 20 µL was pipetted onto freshly cleaved mica and incubated for 1 min. Mica was rinsed with MilliQ water and dried with high purity N_2_ gas. In some experiments, the titin sample was exposed to meniscus forces by using methods published previously [48]. Briefly, the sample was diluted in chromatography buffer supplemented with 50% glycerol. 20 µL of the sample was pipetted onto freshly-cleaved mica mounted in a custom-built rotor, which was immediately spun with 13,000 rpm for 20 s. Subsequently, the sample was rinsed with MilliQ water and dried with N_2_ gas. When imaging in liquid, the sample on the mica substrate was gently washed with 20% PBS. AFM imaging was carried out in this buffer.

### 2.3. Nanosurgical Manipulation

Nanosurgical manipulation was carried out by pushing the tip of an AFM cantilever onto the sample, then moving the tip sideways so as to dislocate targeted parts of the molecular complex of titin [43]. First, a control AFM image was collected, with a Cypher ES scanner (AsylumResearch, Santa Barbara, CA, USA), in tapping mode at a setpoint 60–70% of the cantilever’s free oscillation amplitude so as to prevent unwanted mechanical damage of the sample. Typical scanning rates were 0.4–1.2 Hz. Silicon-nitride cantilevers (OMCL-AC160TS-R3, Olympus, Tokyo, Japan) were employed with a tip radius of 7 nm and nominal spring constant of 27 N/m. Subsequently, the starting point, the direction, the distance and speed of the AFM tip movement were adjusted. The probe was pushed against the surface, in contact mode, with force ranging between 150–600 nN. The speed and distance of probe movement was varied between 10–1000 nm/s and 10–300 nm, respectively. Finally, a second AFM image was collected, with scanning parameters identical to those of the initial scan, to reveal the structural changes caused by the nanosurgical manipulation.

### 2.4. Image Analysis, Processing, and Statistics

Images were processed by the AsylumResearch AR16 software, which is based on Igor Pro 6.34 (WaveMetrics, Lake Oswego, OR, USA). For certain analytical tasks and deconvolution, we used Gwyddion (v.2.60, GNU Public License). Images were corrected for flatness of field and color contrast. Topography was analyzed from section graphs of the manually traced contour of the multimeric sample. Volume was calculated as the integral of the surface height over the masked area. For volume comparison, we used correction methods based on a region containing a well characterizable object. This object was unmanipulated, and we used its volume between two different scans to calculate a ratio for the possible inter-scan bias of volume. With this ratio, the manipulated volumes were corrected. For plotting and analysis of numerical data, we used Microsoft Excel 2016 (Microsoft, Redmond, WA, USA) and Prism 9 (GraphPad, San Diego, CA, USA).

## 3. Results and Discussion

### 3.1. Topographical Structure, Stability, and Organization of the Titin M-Complex

In this work, we used nanosurgical manipulation to explore the structural, mechanical, and material properties of the globular head-like structure at the C-terminal end of the giant muscle protein titin, which we purified from rabbit back muscle (Figure 1a). Considering that the C-terminus of titin extends into the M-band of the muscle sarcomere [17] where titin interacts with a number of proteins such as myomesin [21], M-line protein [21], myomesin-3 [49], and obscurin [50], we hereby refer to the globular head-like structure of titin as the M-complex, even though in the present work, we did not directly quantify the proportion of the additional protein components. The M-complex serves as a focal point of interaction in head-to-head oligomers of titin (Figure 1b) [17,47,51,52]. In a three-dimensional space, the M-band is a thick, two-dimensional plate traversing the cross-section of the myofibril and positioned in the center of each sarcomere [53]. Accordingly, the M-complex appears to be a separable structural unit of the M-band into which titin molecules from either side of the sarcomere are projected. The number of titins projecting into an M-complex may vary widely (Figure 1c,d). In the central area of M-complexes with a large number of titins, sometimes a thickened, confluent region appeared (Figure 1c), pointing at a tendency of interaction between the titin molecules in this area. Conceivably, this interaction is related to the pairwise organization of titin molecules on the surface of myosin thick filaments [54]. In these large M-complexes, a subunit structure could be revealed by using phase-contrast AFM imaging (Figure 1(c.iii)). The subunit structure is possibly related to the appearance of M-lines within the M-band [18,19]. As the histogram of the number of titin molecules has local modes at 6 and 12 (Figure 1d) [47,51], the M-complex is most likely a structural unit that reflects the transversal organization of titin molecules dictated by the myosin thick filament, even though myosin molecules, which have a characteristic two-headed appearance in AFM images [47], were totally absent from the sample. The complete absence of myosin is likely due to efficient chromatography purification. We observed a strong linear correlation between the number of titin molecules and the volume of the M-complex (Figure 1e). As the volume-axis intercept of the fitted linear function is 0, the formation of the M-complex is unlikely in the absence of titin, and the titin-associated molecular unit contributes with a 1–1 stoichiometry to M-complex formation. The resolvable contour length of titin molecules within the titin oligomer was significantly smaller than that of single molecules (Figure 1f), which further points at the possibility of structural compaction near the M-complex due to lateral interaction between the individual titin strands. We tested the stability of the M-complex by treating titin oligomers with 4 M urea (Figure 1g). Interestingly, large oligomers with structural arrangement indistinguishable from that in the control sample were observed. The finding indicates that the M-complex is a highly stable structure, which cannot be dissociated with chaotropic agents or high ionic strength (used during the titin purification procedure), even though the globular head of titin can be digested away with proteolytic treatment [17]. Altogether, the titin M-complex is a stable structure that reflects the transverse unit of the M-band organized around the myosin thick filament.

### 3.2. Architecture of the M-Complex

In many large titin oligomers, the central region appeared as a string or array of several globular heads (Figure 2(a.i,a.ii)) separated by an average distance of ~30 nm (29.37 ± 7.12 nm SD, *n* = 20) (Figure 2(a.ii.profile)). The plausible explanation for the emergence of such an M-complex array is that neighboring M-complexes from the M-band lattice stay attached during the breakage of the M-band during titin preparation (Figure 2(a.iii)). The average distance between neighboring M-complexes within the array measured here (~30 nm) is smaller than the thick-filament spacing measured in different muscle types at physiological ionic strength and osmotic concentration (~40 nm) [55,56,57]. There are two factors that likely contribute to the shortened inter-M-complex distance. First, at the high ionic strength of the sample buffer (~0.6 M, chromatography buffer), the surface charges of component filaments are electrostatically screened and the thick-filament lattice spacing becomes reduced [57]. Second, the myomesin molecules interconnecting the M-complexes within the M-band [15,22] are highly elastic [25,58,59], hence able to pull vicinal M-complexes closer together in the mechanically relaxed configuration of the isolated M-complex array. The large variation of the inter-M-complex distance observed in our measurements (~15–45 nm, Figure 2(a.ii.profile) inset) may be explained by the rupture of myomesin dimers in between neighboring M-complexes, leading to the rearrangement of the M-complexes and the development of uneven inter-M-complex distances (Figure 2b). Occasionally, we observed broken M-complex arrays (Figure 2c), which lends support to this notion. Furthermore, positional instability, caused by ruptured M-band, has been observed before in activated skeletal muscle sarcomeres [60], lending further support to our explanation. We note, however, that the myomesin-dependent explanation detailed here is somewhat speculative, as the exact amount of myomesin within the specific M-complexes are not known. In some titin oligomers, the individual titin strands radiating into/out of the M-complex array was discernible (Figure 2d), allowing us to measure the titin-filament spacing (Figure 2(d.profile)). The mean titin-filament spacing was 20.14 nm (±7.12 nm SD, *n* = 14), which compares well with the expected spacing between titin quadruplexes in the M-band cross-section [15,18,22]. This observation provides further support for the idea that the M-complex reflects the position of the myosin thick filament in the M-band lattice, rather than any other structure with spatial periodicity (e.g., M-filaments [19]). Altogether, M-complexes may be interlinked (by elastic myomesin dimers [18]) into an M-complex array, which reflects the local structural and mechanical status of the M-band lattice from which they were broken out.

### 3.3. Nanosurgical Manipulation of Titin Molecules

We started the nanosurgical manipulation of titin oligomers by manipulating the titin filaments radiating into/out of the M-complex. First, the speed of AFM tip displacement was tested (Figure 3). The titin filaments behaved differently when manipulated at different velocities. Figure 3(a.i,a.ii) shows the effects of nanosurgical manipulation with a low, 10 nm/s speed. The AFM probe movement direction was perpendicular to the orientation of the titin filament to be manipulated. A segment of the titin filament became displaced, stretched, and eventually ruptured at the farthest point. Considering that the stretched titin strands are clearly visible on both sides of the tip path, the manipulated molecular segment became stabilized on the substrate surface during or immediately following the nanosurgical manipulation. By comparing the cross-sectional topographical height profiles of the extended strands and the initial titin filament segment (Figure 3(a.i.profile,a.ii.profile), we found that the diameter of the titin molecule was reduced (from ~0.3 to 0.2 nm), indicating that structural changes, possibly partial domain unfolding, occurred. Nanosurgical manipulation of titin molecules with an AFM tip moving with a speed of 1000 nm/s is shown in Figure 3(b.i,b.ii). While the overall shape changes in the titin molecule are similar to those seen after manipulation with a speed of 10 nm/s, there are quantitative differences. When manipulating with a speed of 10 nm/s, a longer segment of the molecule became displaced, and the relative extension (i.e., strain), expressed as the (L_1_ + L_2_)/L_0_ ratio (where L_0_ is the initial length of the displaced segment, and L_1_ and L_2_ are the lengths of the two stretched titin strands), of the molecule was approximately 4 (Figure 3(c.i)). By contrast, when manipulating with 1000 nm/s, the strain was about 6 (Figure 3(c.ii)). The differences most likely reflect the relationship between the pulling speed and the factors that determine the topology of titin on the substrate. The topology is determined by titin elasticity, the kinetics of domain unfolding at the instantaneous force, and the equilibrium of interactions, under the specific load, that hold the molecule on the surface. Thus, the fate of nanosurgical manipulation and the ultimate topology of the molecule on the substrate are governed by a competition between (i.e., relative rates of) the AFM tip movement and the intramolecular and surface-molecule interactions. Conceivably, at high AFM tip speeds, only a short segment (i.e., small native contour) of titin becomes dislodged from the substrate because the rate of the dissociation of the bonds holding the molecule on the surface is not high enough to compete with the high speed of the manipulation. Although only a short segment is being pulled by the tip, due to the rapid tip movement the intramolecular stress, the strain [61] becomes large prior to the stabilization of the molecule on the surface. By contrast, at low manipulation speeds, a larger segment (i.e., longer native contour) of titin becomes dislodged from the substrate, because the bonds holding the molecule on the surface have sufficient time to progressively dissociate. However, under these conditions, the intramolecular stress, hence the strain, remains relatively low because these parameters are distributed over a larger contour of the titin filament (Figure 3(c.iii)). In the end, the lengths of the extended strands are comparable regardless of the pulling speed, even though the stress and strain within the strands are different. In these experiments, the length of the segments was distributed around 65 nm (65.61 ± 18.97 nm S.D., *n* = 28) (Figure 3d). In Figure 3e, we show a schematic view of the intramolecular structural changes that may occur as a result of stretching a segment of titin with nanosurgical manipulation. Considering that titin is a linear chain of serially linked ß-barrel globular (Ig- or FN-type) domains interspersed with unique sequences (e.g., PEVK domain) [1], the hierarchy of events is as follows: straightening of the domains, domain unfolding, eventual breaking of the protein chain [62]. Whether the complete breakage of the protein chain indeed occurs, or a completely unfolded and fully extended domain is present in the apex of the stretched titin loop is difficult to discern, because such domains do not display good enough height contrast in AFM images [48]. It is worth noting that considering the dimensions of a globular titin domain (4 nm [63] plus linker region) and that domain unfolding events extend titin in ~28 nm steps [64], nanosurgical manipulation of titin with a speed of 1000 nm/s may actually result in the essentially complete unfolding and extension of the dislodged molecular segment (see Figure 3(c.ii): 6-fold extension of a segment that contained ~5 domains). Notably, the strands of the nanosurgically manipulated titin-molecule segments display a non-linear shape, which may be modeled with an equation derived from the theory of displacement of two-dimensional plates due to in-plane stress and adapted to fibers [37] (Figure 3f):(1)$$u\left(r\right)=k\cdot \ln\left(\left|\frac{r_{0}}{r}\right|\right)$$
where *u*(*r*) is titin-strand extension as a function of distance *r* along the original contour of the molecule from *r*_0_ (the starting point of molecule displacement) where *r* is zero and *u(r)* is maximum, and *k* is
(2)$$k=\frac{\left(1+v\right)P_{x}}{4\pi E}\left(3-v\right)$$
where *v* is the Poisson ratio of the material, *P_x_* is applied tension (in N/m units), and *E* is Young’s modulus. We carried out a least-squares fit to the raw data obtained by superimposing a *u*(*r*) versus *r* coordinate system on the AFM image of a nanosurgically manipulated titin loop (Figure 3(f.i)). The results of the fit are shown in Figure 3(f.ii) (blue trace). The original model fails to provide an optimal fit due to the formation of a gap between the strands at the vertex of the molecular loop, which is caused by the AFM tip itself. By incorporating the AFM tip radius as a shift factor *d* into the model, we obtain a new set of equations:(3)$$u\left(r\right)=\{\begin{array}{c} k\cdot \ln\left(\frac{-r_{0}}{r+d}\right),\;if\;r<0\\ k\cdot \ln\left(\frac{r_{0}}{r-d}\right),\;if\;r>0 \end{array}$$
which provide a significantly better fit to our nanosurgical manipulation results (Figure 3(f.ii), red trace). In sum, distinct segments of the titin molecule can be nanosurgically manipulated to achieve local extension and domain unfolding, and the shape of the stretched titin strands may be described by a modified thin-plate model.

### 3.4. Nanosurgical Manipulation of the M-Complex

We carried out nanosurgery on the titin M-complex at several levels of manipulation complexity. First, we tested the effect of cutting through the center of the M-complex with two different AFM-tip movement speeds (10 versus 1000 nm/s) (Figure 4a–c). We found that a slower tip movement resulted in a more thorough cut across the M-complex (Figure 4b insets). The finding may be explained with the same mechanisms as in the case of titin filament manipulation: at high AFM tip speeds, only a small portion of the M-complex became dislodged because the equilibrium of interactions holding the complex intact was not fast enough to compete with the high speed of the manipulation. By contrast, at low AFM tip speed, a larger portion of the M-complex became dislodged, and a more complete molecular rearrangement was permitted by the slower nanosurgical manipulation rate. Therefore, in order to carry out more thorough nanosurgical manipulations, we used slow (10 nm/s) AFM tip speeds in the subsequent experiments. In a second level of nanosurgical manipulation complexity, we carried out two AFM-tip movements from the same starting point in the center of the M-complex, but in directions perpendicular to each other (Figure 4(d.i,d.ii). Although these experiments were prone to debris deposition (see Figure 4(d.ii) insets 2 and 3), it is clear that loops of filaments could be pulled out of the M-complex in every direction of the manipulation, suggesting that there is a reservoir of filaments in the titin M-complex. However, assessing the features of the stretched strands was complicated by the presence of the densely packed region of titin molecules radiating into/out of the M-complex. To alleviate this problem, we extended the titin oligomer by meniscus forces [48] prior to nanosurgical manipulation (Figure 4e). In titin oligomers exposed to meniscus forces, the titin filaments became oriented along the direction of the centrifugal force [48] (Figure 4(e.i)). By manipulating the M-complex in directions perpendicular (or close to perpendicular) to the axis of the meniscus-stretched titin oligomer, the filament strands pulled out of the M-complex became discernible (Figure 4(e.ii)). We found that the topographical height of the stretched strands decayed with the distance, along the direction of nanosurgical manipulation, from the center of the M-complex (Figure 4(e.iii)). The topographical height as a function of distance (*x*) could be fitted well with a sigmoidal function:(4)$$B+\frac{\left(T-B\right)}{\left[1+{\left(\frac{half\;height}{x}\right)}^{S}\right]}$$
where *B* and *T* are the minimal the maximal heights, respectively, half height is the distance where topographical height equals (*T − B*)/2, and *S* is a slope factor related to the steepness of the curve. The sigmoidal function suggests that, starting from the M-complex center, there is a transition from a bulky to a thinned region upon approaching the end of the pulled-out filamentous strand. The coefficient *S* describes how gradual the transition is. Conceivably, the M-complex is a bulky filament reservoir from which a few strands are pulled out during nanosurgical manipulation, resulting in nematic ordering and increasing structural inhomogeneity and anisotropy. Long filamentous strands with lengths nearing 300 nm could be pulled out of the M-complex (Figure 4f). The observed exponentially decaying distribution of the pulled-out filament length is in accordance with the sigmoidal function of topographical height versus filament length (Figure 4(e.iii)). In sum, the titin M-complex is a compact supramolecular structure that may be seen as a reservoir of extensible filaments.

### 3.5. Manipulation-induced Volume Change of the M-Complex

We measured the changes in the volume of the M-complex evoked by nanosurgical manipulation (Figure 5). The M-complex volumes before and after the manipulation were quantitated by thresholding and segmentation (Figure 5(b.i,b.ii)). A pairwise analysis of the results is shown in Figure 5c. As a result of most of the nanosurgical manipulation experiments, the M-complex volume became increased. The volume increment distribution therefore displays overwhelming positive values which are distributed exponentially (Figure 5c inset). The average volume increment is ~3300 nm^3^ (which, to make didactic comparisons with sarcomeric dimensions, amounts to a sphere with a ~19 nm diameter). The mechanically evoked volume increment suggests that elastic energy was released upon the nanosurgical manipulation, which lead to the geometric expansion of the M-complex. Hence, the M-complex is in a mechanically compressed state initially. We speculate that such a compression may arise during the adsorption of the titin oligomer to the substrate surface, which involves the topological distortion of a three-dimensional structure (i.e., titin filaments from both sides of the sarcomere attached to a common M-band structure) into the two-dimensional plane of the mica surface. The capacity of the M-complex to bear compressive forces supports the idea that myomesin, the main M-band protein, may function as a spacer [15,18,65]. In sum, nanosurgical manipulation caused an unexpected volume increment in the M-complex, which may be related to its function as a mechanical spacer.

### 3.6. Bulk Material Properties of the M-Complex

When manipulating bulkier structures of the M-complex (e.g., M-complex array), we observed necking [66], which manifested in the local thinning of the filamentous structure pulled out of the bulk (Figure 6). In such experiments, a portion of the M-complex array was dislodged by the nanosurgery and pushed away from the initial point of the manipulation (Figure 6(a.ii)). The filamentous structure interconnecting the translocated and stationary M-complex-array components became extended, and its topographical height became reduced in the middle (Figure 6b). Such necking behavior implies that plastic deformation occurred in the interconnecting fiber, possibly due to the dissociation of the extended, originally interlinked component filaments from each other (Figure 6(a.iii)). The exact nature of the molecular rearrangements that occur in M-complex fibers under mechanical yield awaits further exploration, but the unusual elastic properties of myomesin [25,58] may conceivably play an important role. In sum, the M-complex displays both elastic and plastic properties which support the idea that the M-band may be involved in mechanical functions [18] within the muscle sarcomere.

### 3.7. Nanosurgical Manipulation of the Titin Oligomer in Aqueous Buffer

The nanosurgical manipulation experiments conducted above on M-complexes were carried out under dry conditions. To circumvent the problems associated with drying, we conducted nanosurgical manipulation experiments under aqueous buffer conditions (Figure 7). First, we tested the effect of buffer ionic strength on titin oligomer adsorption to the substrate (Figure 7a). Upon reducing ionic strength, the titin oligomers became more firmly attached onto mica, and the individual titin filaments radiating into/out of the M-complex could be resolved. Notably, the mean central topographical height of the M-complex was nearly four times greater in liquid than in air (Figure 7(a.v)) (16.64 ± 2.87 nm versus 4.39 ± 1.48 nm). By contrast, the titin molecule height was about six times as large in liquid than in air (5.89 ± 1.66 nm versus 0.97 ± 0.13 nm). However, considering the M-complex as a sphere and the titin filament as a cylinder, and that the height values correspond to the diameter of these geometrical objects, the volume increment in liquid was ~55-fold versus ~37-fold for the M-complex versus titin, respectively. We speculate that the greater volume increment in the M-complex may be related to the release of compressional elastic energy, just as it occurred upon mechanical manipulation. In the rest of the experiments, we carried out the nanosurgical manipulation in buffer containing 20 mM KCl (Figure 7b–c). We observed that larger segments and pieces of the titin oligomer could be dislodged from the mica surface. However, the M-complex could be deformed in these experiments as well, further supporting prior notions that it is an elastic supramolecular structure (Figure 7b). When pulling an entire titin oligomer through its M-complex (Figure 7c), we observed that the titin filaments remained interconnected via the M-complex, indicating that the M-complex maintains cohesion against mechanical force. Altogether, the nanosurgical manipulation experiments conducted in liquid support our findings in the experiments conducted in air.

## 4. Conclusions

In conclusion, high-resolution AFM measurements indicate that the titin M-complex is a stable structure that corresponds to the transverse unit of the M-band organized around the myosin thick filament. M-complexes may be interlinked, most likely by elastic myomesin dimers, into an M-complex array, which reflects the local structural and mechanical status of the transversal M-band lattice. Distinct molecular segments of titin, stemming from the M-complex, could be nanosurgically manipulated to achieve local extension and domain unfolding. The shape of the stretched titin strands may be described by a modified thin-plate model. The titin M-complex appears to be a compact supramolecular structure that may be a reservoir of extensible filaments. Nanosurgery evoked an unexpected volume increment in the M-complex, which may be related to its function as a mechanical spacer. The M-complex thus displays both elastic and plastic properties, which support the idea that the M-band may be involved in mechanical functions within the muscle sarcomere. The nanosurgical manipulation methods employed here may be utilized in the exploration of structure and mechanics of complex molecular assemblies.

## Figures and Tables

**Figure 1 nanomaterials-12-00178-f001:**
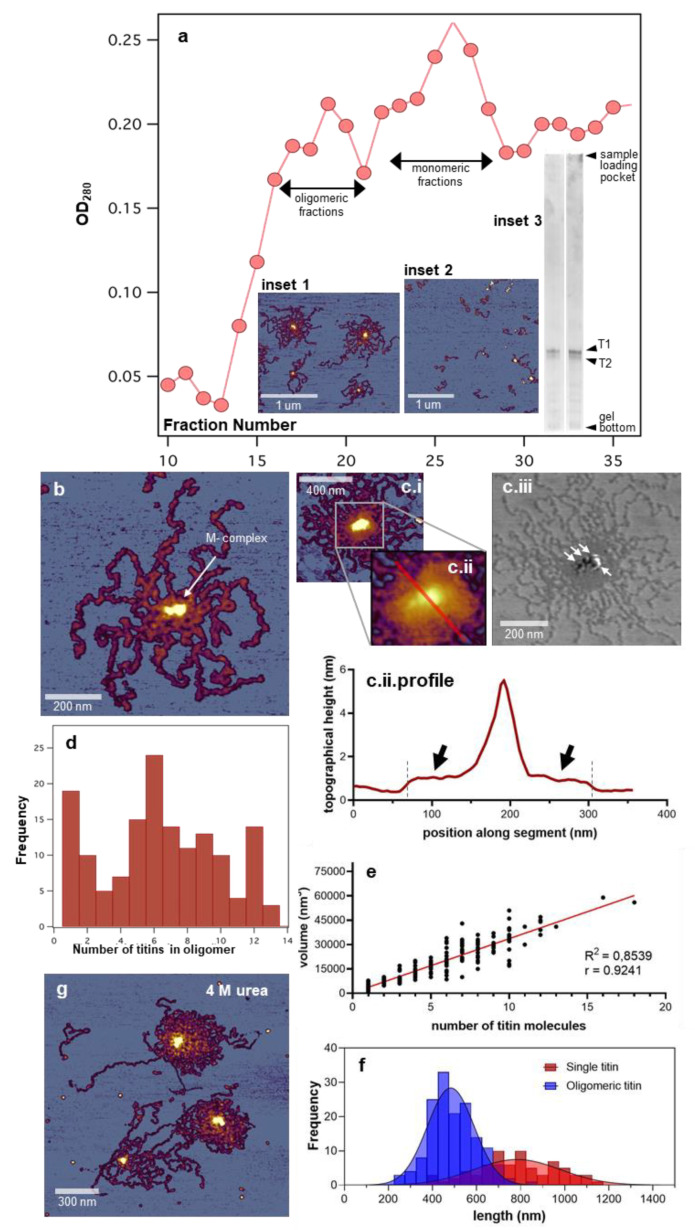
Preparation, overall structure, and stability of the titin M-complex. (**a**). Sepharose CL-2B gel filtration elution profile of our titin sample. (**Inset 1**,**Inset 2**) show AFM images of oligomeric and monomeric fractions, respectively. (**Inset 3**) shows the SDS agarose (0.8 %) gel profiles of purified titin samples. Left and right lanes are representative for oligomeric and monomeric fractions, respectively. (**b**). AFM image of a typical titin oligomer. The arrow points at the M-complex, which is the focal point of head-to-head association between the component titin molecules. (**c.i**). AFM image of a titin oligomer comprising a large number of titin molecules. (**c.ii**). Enlarged view of the middle region of the titin oligomer. Red line marks the section used for obtaining a topographical profile along the center of the M-complex. (**c.ii.profile**). Topographical height profile along the line in c.ii. The arrows point at a plateau region that appears around the M-complex. The dashed lines indicate the edges of the plateau region. (**c.iii**). Phase-contrast AFM image of the titin oligomer shown in c.i. Arrows point at resolvable M-complex subunits. (**d**). Histogram of the number of titin monomers in the oligomer. Local modes appear at 6 and 12. (**e**). Volume of the M-complex, measured from AFM data, as a function of the number of titin molecules in the oligomer. (**f**). Distribution of the resolvable contour length of titin in monomers (“single titin”) and oligomers. The r^2^ values for the gaussian fits to the oligomeric and single-titin data are 0.85 and 0.47, respectively. (**g**). Height-contrast AFM image of titin oligomers in 4 M urea.

**Figure 2 nanomaterials-12-00178-f002:**
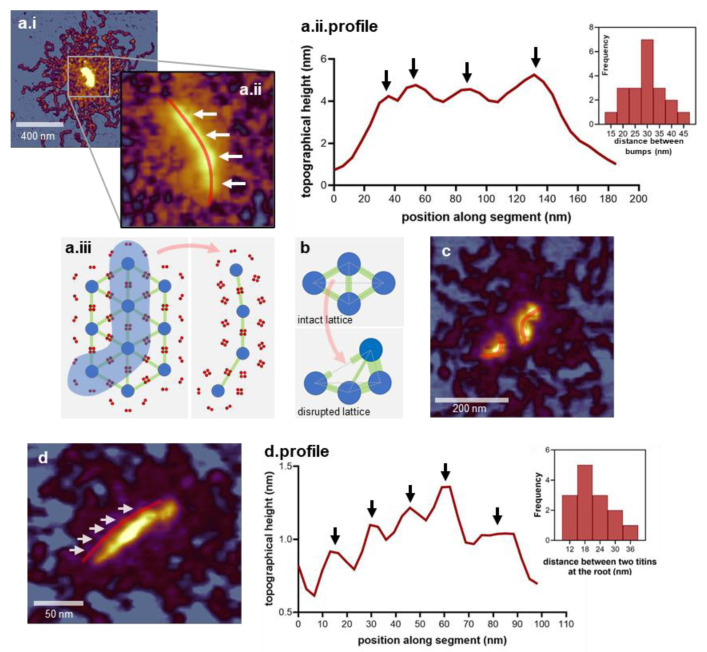
Supramolecular architecture of the M-complex. (**a.i.**) Height-contrast AFM image of a large titin oligomer displaying an array-like M-complex. (**a.ii**). Enlarged view of the M-complex array. Arrows point at the topographical “bumps” identified as individual M-complexes. The red line running through the center of the M-complexes marks the section along which a topographical profile was obtained. (**a.ii.profile**). Topographical height profile along the M-complex array. Arrows mark the topographical peaks corresponding to M-complex centers. (**Inset**), distribution of distance between neighboring height peaks in the M-complex array. (**a.iii**). Schematic model of obtaining the M-complex array. Part of the sarcomeric M-band lattice, highlighted with light blue, is preserved during purification. Dark blue: myosin thick filaments in cross-section; green: M-bridges containing myomesin; red: two pairs of titins from each side of the sarcomere. (**b**). Schematic model of the mechanical disruption of the M-band lattice leading to a wide distribution of the distance between neighboring M-complexes. The thickness of the interconnecting green lines indicates the contractile state of the M-bridges (i.e., thick: contracted; thin: extended). (**c**). AFM image of an M-complex array dissociated into subunits. (**d**). AFM image of an M-complex array. Arrows point at the titin filaments entering the M-complex array. The red line marks the section along which a topographical profile was obtained. (**d.profile**). Topographical profile plot along the titin filaments entering the M-complex array. Arrows mark the topographical peaks corresponding to the titin filament axes. (**Inset**) shows the distribution of distance between neighboring topographical peaks.

**Figure 3 nanomaterials-12-00178-f003:**
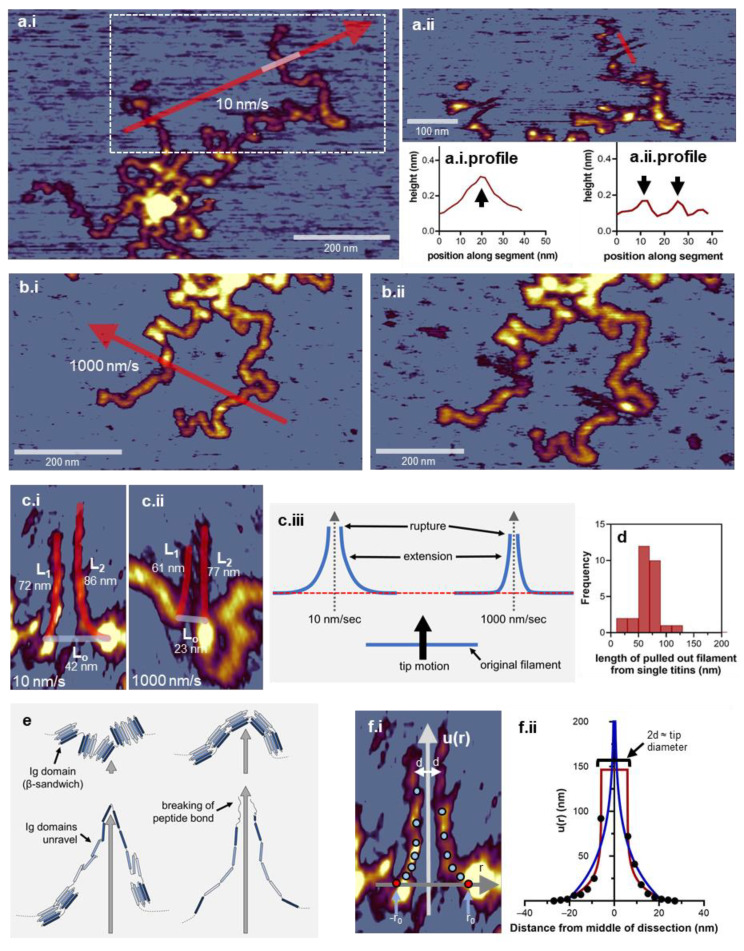
Nanosurgical manipulation of titin molecules. (**a.i**). Control AFM image of a titin oligomer prior to manipulation. Arrow shows the direction AFM tip movement with a speed of 10 nm/s. The bright section of the arrow indicates the area used for topographical height analysis. (**a.i.profile**). Topographical height profile along the cross-section of a titin filament (shown in (**a.i**)). (**a.ii**). Enlarged view of the AFM image of the manipulated titin oligomer. Red line indicates the area used for topographical height analysis. (**a.ii.profile**). Topographical height profile along the cross-section of a nanosurgically pulled titin filament (shown in (**a.ii**)). The two height-profile maxima correspond to the two strands of the pulled titin loop. (**b.i**). Control AFM image of a titin oligomer prior to manipulation. Arrow shows the direction AFM tip movement with a speed of 1000 nm/s. (**b.ii**). AFM image of the manipulated titin oligomer. (**c.**) Effect of manipulation speed on titin filament structure. (**c.i**,**c.ii**). Enlarged AFM images of titin filaments pulled with speeds of 10 and 1000 nm/s, respectively. L_0_ indicates the segment of titin that became stretched out. L_1_ and L_2_ indicate the strands of the pulled-out titin loop. (**c.i****ii**). Schematic model of the titin-segment nanosurgical manipulation with different speeds. (**d**). Distribution of the length of the titin-loop segments pulled out from single titin molecules. (**e**). Schematic steps of the molecular events underlying the stretching of the titin filament with nanosurgical manipulation. For the sake of simplicity, only five out of the seven ß-strands of titin’s domains are shown. (**f.i**) Enlarged view of a filament loop pulled out of titin with a superimposed coordinate system for analysis. The vertical arrow shows the position, direction, and length scale (*u*) of nanosurgical manipulation, the horizontal arrow is the tangent vector positioned on the starting point of the manipulation, *r*_0_ is the radius of the initial loop, and *d* is the AFM tip radius so that 2*d* is tip diameter. Circles mark the image points for distance measurements. (**f.ii**). Local loop extension (*u*) as a function of distance, along the original contour of titin, from the initial point of manipulation. Blue curve is the fit of the original thin-plate model to our data. For fitting, the coefficients *P_x_* (applied force), *E* (Young’s modulus) and *v* (Poisson ratio) were merged in a single coefficient *k* according to equation 3. We note that the model can be used if the shape of the displacement is nearly symmetrical to the *y* axis.

**Figure 4 nanomaterials-12-00178-f004:**
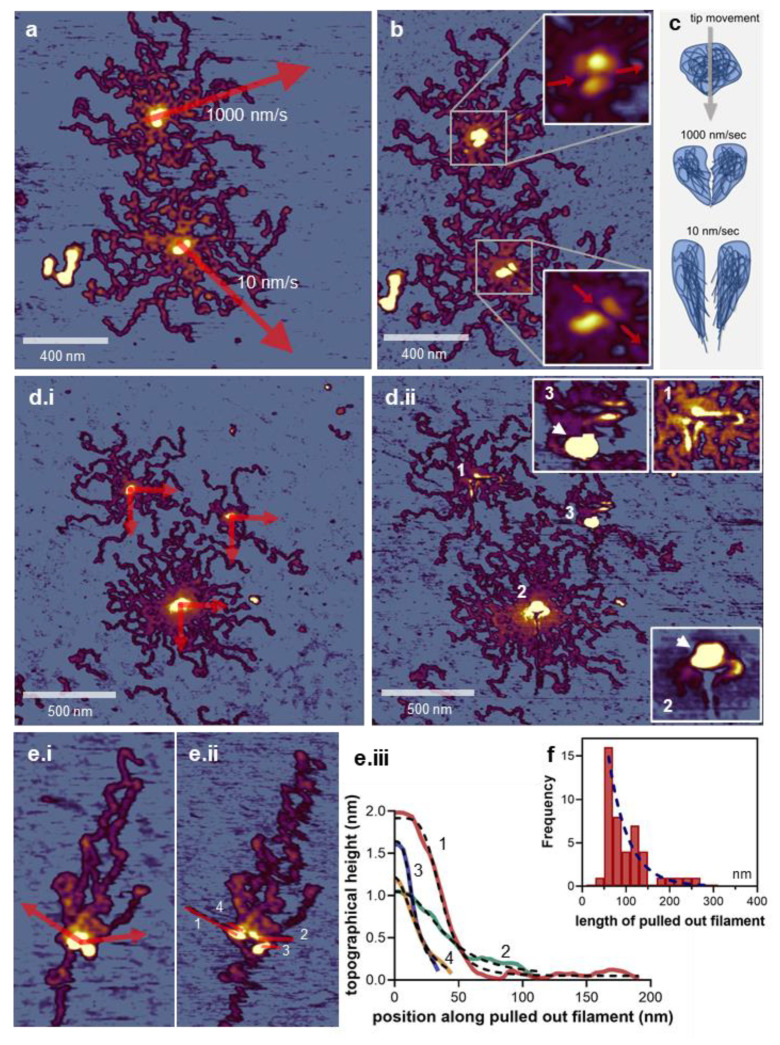
Nanosurgical manipulation of the titin M-complex. (**a**). AFM image of two titin oligomers prior to nanosurgical manipulation. Arrows indicate the site, direction, and distance of manipulation at the indicated speeds (10 and 1000 nm/s). (**b**). AFM image recorded after the nanosurgical manipulation. (**Insets**) show enlarged views of the two M-complexes. (**c**). Schematics explaining the effects of nanosurgical manipulation of the M-complex at different manipulation speeds. (**d.i**). Control AFM image of three titin oligomers containing M-complexes of different size prior to nanosurgical manipulation. Arrows indicate the site, direction, and distance of manipulation. (**d.ii**). AFM image recorded after the nanosurgical manipulation. (**Insets 1**–**3**) show the magnified images of each modified structure. Arrows point at regions of material accumulation, possibly due to debris deposition. (**e.i**). AFM image of a titin oligomer stretched with meniscus force prior to nanosurgical manipulation. Arrows indicate the site, direction and distance of manipulation. (**e.ii**). AFM image recorded after the nanosurgical manipulation. Numbers indicate the filaments pulled out of the M-complex, which were further analyzed for axial height distribution. (**e.ii****i**). Topographical height of the pulled-out filament as a function of distance from origin. Numbers correspond to the filaments marked in e.ii. Segmented lines are fits of equation 1 on the data. The r^2^ values of the fits are >0.95 for each dataset. (**f**). Distribution of the length of filaments pulled out of the M-complex. The distribution was fitted with the exponential function *f* = e*^−cl^*, where *f* is statistical frequency, *l* is filament length, and *c* is a characteristic length scale. The r^2^ value of the exponential fit is 0.87.

**Figure 5 nanomaterials-12-00178-f005:**
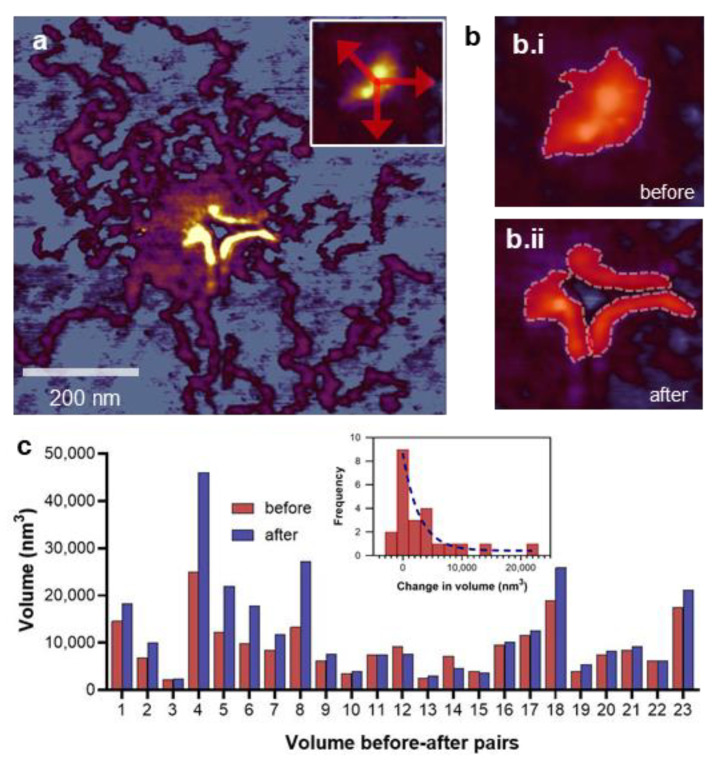
Volume changes in the nanosurgically manipulated M-complex. (**a**). AFM image of a M-complex nanosurgically manipulated in three different directions (indicated by red arrows in the inset). (**b.i**,**b.ii**): Method of comparing the a priori and a posteriori volumes of the M-complex by thresholding and masking. Red areas show the masked regions for volume analysis. (**c**). Pairwise analysis of the a priori and a posteriori M-complex volumes. (**Inset**), distribution of the change in volume of the M-complex as a result of nanosurgical manipulation. The distribution was fitted with the exponential function *f* = e^−*w*Δ*V*^, where *f* is statistical frequency, Δ*V* is volume change and *w* is the characteristic scale of mechanically evoked volume change in the M-complex (~30,000 nm^3^). The r^2^ value of the exponential fit is 0.91.

**Figure 6 nanomaterials-12-00178-f006:**
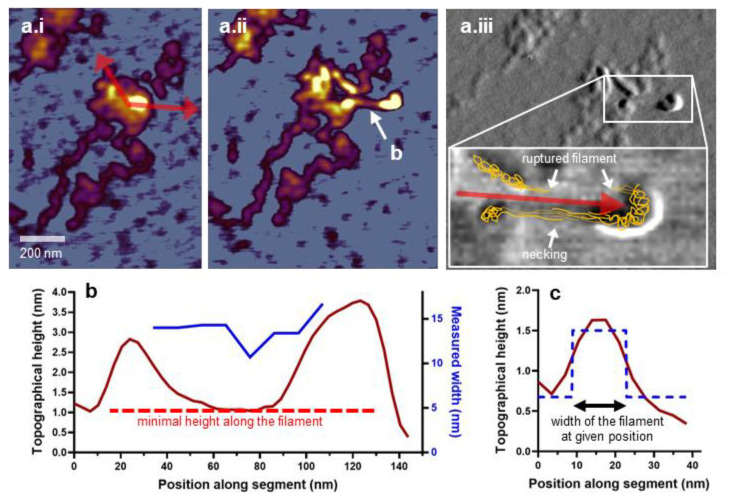
Ductility and necking in the nanosurgically manipulated M-complex. (**a.i**) Height-contrast AFM image of a titin oligomer before manipulation. Arrows indicate the site, direction, and distance of the manipulation. (**a.ii**) AFM image recorded after the nanosurgical manipulation. White arrow points at the titin strand analyzed for topographical height and width. (**a.iii**) Phase-contrast AFM image of the manipulated M-complex. The boxed area is enlarged in the inset and supplemented with a schematic drawing of the likely molecular arrangement in the pulled titin loop. The red arrow indicates the AFM tip motion during nanosurgical manipulation. Above the arrow, the titin strand has ruptured. By contrast, below the arrow, the titin strand has gone through necking, which entails the thinning of the structure in the middle. (**b**) Topographical analysis of the necked titin strand. Red and blue curves are the topographical height and width of the titin strand as a function of axial position along its length, respectively. (**c**) Method of calculating the width of the necked titin strand, by fitting a square function to its cross-sectional topographical profile.

**Figure 7 nanomaterials-12-00178-f007:**
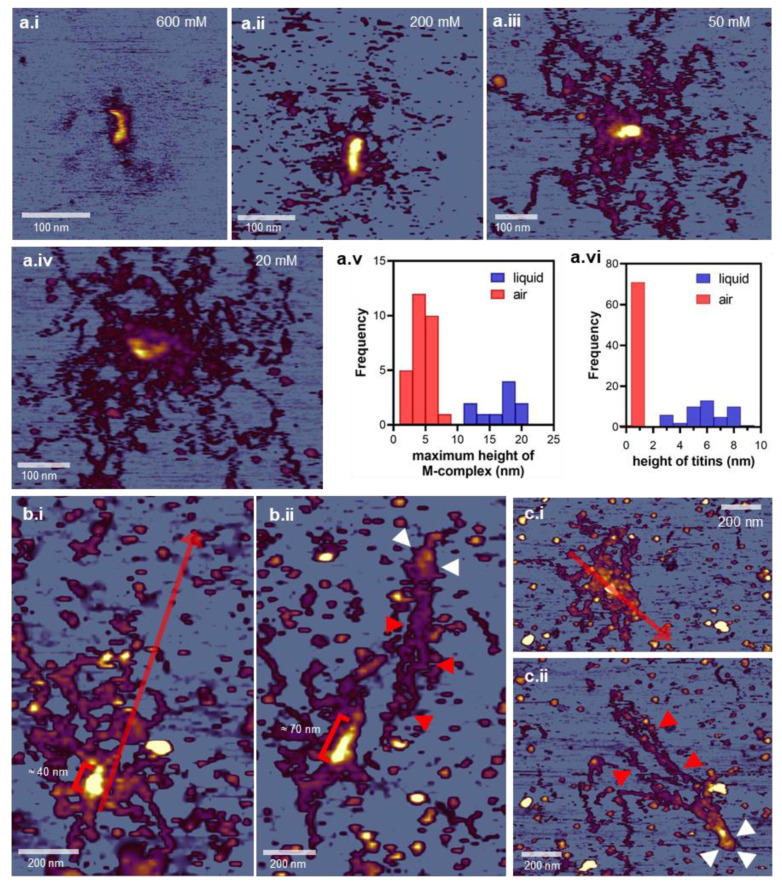
Nanosurgical manipulation of the M-complex in aqueous buffer solution. (**a**) Effect of ionic strength on the adsorption of the titin oligomer to mica. (**a.i**–**a.iv**) AFM images of titin oligomers at progressively decreasing ionic strengths. The ionic strength was adjusted with KCl, the concentration of which is indicated in the upper right corner of the images. (**a.v**) Comparison of the maximum topographical height of the M-complex in air (red) and aqueous buffer (blue). (**a.vi**) Comparison of the maximum topographical height of titin molecules in air (red) and aqueous buffer (blue). (**b.i**) Control AFM image of a titin oligomer before nanosurgical manipulation in buffer. Red arrow indicates the site, direction and distance of the manipulation. The red bracket marks the M-complex. (**b.ii**) AFM image recorded after nanosurgical manipulation. Besides extension (from 40 to 70 nm), large parts of the titin oligomer were dislodged and displaced. Red and white arrowheads mark displaced titin strands and parts of the M-complex, respectively. (**c.i**) Control AFM image of a titin oligomer before nanosurgical manipulation in buffer. Red arrow indicates the site, direction, and distance of the manipulation. (**c.ii**) AFM image recorded after nanosurgical manipulation. The entire titin oligomer was displaced with a net distance of 750 nm. Red and white arrowheads mark displaced titin strands and parts of the M-complex, respectively.

## Data Availability

Data supporting the reported results has been uploaded to the institutional server and is made available upon request to the corresponding author.

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
