# Peer review of "Nanosurgical Manipulation of Titin and Its M-Complex"

_nanomaterials, 2022, doi:10.3390/nano12020178_

Round 1
Reviewer 1 Report
This study reports on the biomechanical properties of titin M-line investigated by atomic force microscopy. Authors report unexpected volume increment of the M-line complex upon the active push of the cantilever through the protein complex. The presented manuscript is an innovative and exciting analysis of titin filament.
Major concerns
- Authors should address the protein composition of the M-band preparation experimentally, or otherwise, emphasize the speculative nature of their conclusions about the supramolecular architecture of the M-complex.
- Authors should discuss and/or provide data that observed complexes are not the result of non-physiological aggregation of titin during preparation.
Suggestions
- Provide experimental details for titin-M line complex gelfiltration.
- The quality of curve fitting should be quantified and reported.
- Authors should compare measured titin filament biomechanical properties with those from the work of other authors.
Author Response
Response to Reviewer 1:
Comment:
This study reports on the biomechanical properties of titin M-line investigated by atomic force microscopy. Authors report unexpected volume increment of the M-line complex upon the active push of the cantilever through the protein complex. The presented manuscript is an innovative and exciting analysis of titin filament.
Response: We thank the reviewer for the encouraging and supportive comments.
Major concerns
- Authors should address the protein composition of the M-band preparation experimentally, or otherwise, emphasize the speculative nature of their conclusions about the supramolecular architecture of the M-complex.
Response: We would like to emphasize that the preparations investigated in this work were not M-band preparations per se, but byproducts of the titin preparation itself. Nevertheless, we agree with the reviewer that due to the lack of a detailed analysis of the protein composition of the purified titin samples, we must be cautious by making reference to the speculative aspect of our conclusion with regards to the actual protein participants. We amended the revised manuscript with a gel profile (Figure 1.a inset 3), which shows the bands corresponding to T1 (full-length titin) and T2 (proteolytic product of titin. Although titin is detected in these gels, the small-quantity components of the M-complex were not resolved at this level of sensitivity. Therefore, we amended the Discussion section of the revised manuscript to reflect the speculative aspects, and, in addition, make reference to prior publications in which the globular hear of the titin preparation has been identified with biochemical detail. - Authors should discuss and/or provide data that observed complexes are not the result of non-physiological aggregation of titin during preparation.
Response: Thank you for pointing out the possibility of an artificial aggregation of titin. We note that during the preparation of titin there is one step in which titin is precipitated by lowering the ionic strength, but then the precipitate id dissolved by the addition of 0.6 M KCl, thereby reversing the precipitation effect. Considering that the head-to-head binding of titin is structurally specific (i.e., it does not occur randomly along the molecule's contour), the complexes indeed appear to reflect the in situ arrangements. Because such head-to-head titin oligomers have been seen by others, there appears to be a consensus in the field that they are not artifactual but reflect titin's charateristics.
Suggestions
- Provide experimental details for titin-M line complex gelfiltration.
- The quality of curve fitting should be quantified and reported.
- Authors should compare measured titin filament biomechanical properties with those from the work of other authors.
Response: Thank your for the suggestions. In the revised manuscript we briefly addressed each by making amendments in the Materials and Methods and the Results and Discussion sections. We provided further detail about hte gel filtration step of titin purification; r2 values of the fits were added to the legends of the corresponding figures where applicable; further references were added on myomesin biomechanics in the Discussion section.
Reviewer 2 Report
In the manuscript # nanomaterials-1516242 by Sziklai et al., " Nanosurgical manipulation of titin and its M-complex", authors demonstrated that the titin M-complex was a stable structure that corresponds to the transverse unit of the M-band organized around the myosin thick filament. Furthermore, authors showed successfully that the shape of the stretched titin strands might be described by a modified thin-plate model. Although I have no serious criticisms regarding methodology, results and interpretation of results. A minor revision is listed below.
Minors:
If possible, authors had better provide the pictorial and visual summary of the main findings of the article.
Author Response
Response to Reviewer 2:
Comment:
In the manuscript # nanomaterials-1516242 by Sziklai et al., " Nanosurgical manipulation of titin and its M-complex", authors demonstrated that the titin M-complex was a stable structure that corresponds to the transverse unit of the M-band organized around the myosin thick filament. Furthermore, authors showed successfully that the shape of the stretched titin strands might be described by a modified thin-plate model. Although I have no serious criticisms regarding methodology, results and interpretation of results. A minor revision is listed below.
Response: We thank the reviewer for the encouraging and supportive comments.
Minors:
If possible, authors had better provide the pictorial and visual summary of the main findings of the article.
Response: In the revised manuscript we added a new, expanded graphical abstract to better reflect the main findings of our experiments.
Reviewer 3 Report
This is an interesting study that provides new insights into the nature and structure of titin in the M-band using a range of molecular force techniques. The authors isolate "M-complexes" that are constructed from the C-terminal end of titin and other associated proteins and characterize their mechanical properties. This is a well-designed study that utilizes novel techniques like nanosurgery to demonstrate that the M-complex has both elastic and plastic properties that contribute to the overall mechanical properties of the M-band in the sarcomere. Overall, this is a well-written manuscript and worthy of publication, though there are a few revisions that should be considered.
- On page 3, the sentence starting on Line 141 describes that local number of titin molecules that are observed in the M-complex but it is pointed out that there are no myosin molecules observed in the complex. This seems like an interesting finding that potentially warrants some additional discussion. The authors point out that the M-complex most likely reflects the organization of the titin dictated by the thick filament but it is not clear whether the authors are suggesting that this structure is formed around the thick filament and is stable once formed or if the M-complex forms and provides a template for thick filament formation.
- On page 5, in the architecture of the M-complex section, the authors provide two factors that they believe contribute to the shortening of the inter-M-complex distance, ionic strength and the elasticity of myosesin. However, they don't provide any data to support these hypotheses. For example, you would predict that the distances would return to the predicted distances if the ionic strength was decreased to more physiologically relevant levels. Did the authors try this or was there a technical reason why this was not feasible to do? Also, did the authors do any western blots or other tests to confirm the presence of myomesin?
- On page 7, starting on line 257, the authors discuss the impact of the rate of AFM tip speed and the length of extension of the titin. However, this is a little confusing since the authors talk about how only a short section of titin is dislodged at high speed since the equilibrium of the interactions between the surface and the substrate "is not fast enough to compete with the high speed of manipulation." This part of the sentence could be clarified since it could be misinterpreted that the pulling rate outcompetes the equilibrium stabilizing interactions rather than the alternative.
- On the same page and a few lines down, the authors make the statement that "In the end, the lengths of the extended strands are comparable regardless of pulling speed . . ." This seems contradictory to the statements above in that paragraph that lower pulling rate leads to longer segments being detached from the surface. What the authors mean here should be clarified.
- On page 10, on line 382, the authors model the volume as a sphere with ~19 nm diameter. However, it is not clear why this is a valid assumption for calculating the volume of the complex. The M-complex is presumably sitting on the surface and might be in more of disk shape. There is also a reasonable likelihood that the M-complex would be in a disk shape in the M-band rather than a sphere. It would be helpful for the authors to clarify why it is reasonable to calculate the volume as a sphere instead of a more asymmetrical structure.
Overall, this is an interesting and well-written paper. These suggestions will help clarify a few points.
Author Response
Response to Reviewer 3:
This is an interesting study that provides new insights into the nature and structure of titin in the M-band using a range of molecular force techniques. The authors isolate "M-complexes" that are constructed from the C-terminal end of titin and other associated proteins and characterize their mechanical properties. This is a well-designed study that utilizes novel techniques like nanosurgery to demonstrate that the M-complex has both elastic and plastic properties that contribute to the overall mechanical properties of the M-band in the sarcomere. Overall, this is a well-written manuscript and worthy of publication, though there are a few revisions that should be considered.
Response: We highly appreciate the encouraging and positive comments of the reviewer.
- On page 3, the sentence starting on Line 141 describes that local number of titin molecules that are observed in the M-complex but it is pointed out that there are no myosin molecules observed in the complex. This seems like an interesting finding that potentially warrants some additional discussion. The authors point out that the M-complex most likely reflects the organization of the titin dictated by the thick filament but it is not clear whether the authors are suggesting that this structure is formed around the thick filament and is stable once formed or if the M-complex forms and provides a template for thick filament formation.
Response: Thank you for raising this interesting theoretical issue. Indeed, we did not observe myosin molecules in our preparations, which is likely due to the efficient removal of myosin during the purification procedure. The intimate relationship between titin and the myosin thick filament, particularly in the M-band is, however, an important and not fully resolved question. We speculate that, given the remarkable stability of the M-complex, it does reflect the transverse structure of the M-band, in which the presence of the myosing thick filament plays an important structural role. However, at present it would be too speculative to comment on the sequence of molecular events occurring during the development of the M-band during sarcomerogenesis. - On page 5, in the architecture of the M-complex section, the authors provide two factors that they believe contribute to the shortening of the inter-M-complex distance, ionic strength and the elasticity of myomesin. However, they don't provide any data to support these hypotheses. For example, you would predict that the distances would return to the predicted distances if the ionic strength was decreased to more physiologically relevant levels. Did the authors try this or was there a technical reason why this was not feasible to do? Also, did the authors do any western blots or other tests to confirm the presence of myomesin?
Response: We note here that during sample preparation, the M-complex was exposed to very high ionic strength (0.6 M KCl in the chromatography buffer), hence the notion about the role of ionic strength. Currently we do not have direct evidence for the presence of myomesin in our samples, but we rely on literature data on similar specimens. To demonstrate the biochemical features of the samples tested in this work, we added high-resolution agarose gel profiles in Figure 1 (Figure 1.a inset 3). - On page 7, starting on line 257, the authors discuss the impact of the rate of AFM tip speed and the length of extension of the titin. However, this is a little confusing since the authors talk about how only a short section of titin is dislodged at high speed since the equilibrium of the interactions between the surface and the substrate "is not fast enough to compete with the high speed of manipulation." This part of the sentence could be clarified since it could be misinterpreted that the pulling rate outcompetes the equilibrium stabilizing interactions rather than the alternative.
Response: Thank you for pointing out the potentially misleading nature of the sentence. In the revised manuscript we made due corrections. - On the same page and a few lines down, the authors make the statement that "In the end, the lengths of the extended strands are comparable regardless of pulling speed . . ." This seems contradictory to the statements above in that paragraph that lower pulling rate leads to longer segments being detached from the surface. What the authors mean here should be clarified.
Response: In the revised manuscript we amended the section to increase clarity. - On page 10, on line 382, the authors model the volume as a sphere with ~19 nm diameter. However, it is not clear why this is a valid assumption for calculating the volume of the complex. The M-complex is presumably sitting on the surface and might be in more of disk shape. There is also a reasonable likelihood that the M-complex would be in a disk shape in the M-band rather than a sphere. It would be helpful for the authors to clarify why it is reasonable to calculate the volume as a sphere instead of a more asymmetrical structure.
Response: Thank you for the critical comments about the volume calculation. Comparison of the volume increment to the volume of a sphere was made solely for didactic purposes, so that single-axis dimensions (i.e., diameter) could be related to the molecular dimensions of the M-band. In the revised manuscript we added an explanatory sentence to clarify this issue.
Overall, this is an interesting and well-written paper. These suggestions will help clarify a few points.